# Microbiota Community Structure and Interaction Networks within *Dermacentor silvarum*, *Ixodes persulcatus*, and *Haemaphysalis concinna*

**DOI:** 10.3390/ani12233237

**Published:** 2022-11-22

**Authors:** Hongyu Qiu, Qingbo Lv, Qiaocheng Chang, Hao Ju, Tingting Wu, Shunshuai Liu, Xiuwen Li, Yimeng Yan, Junfeng Gao, Chunren Wang

**Affiliations:** 1Key Laboratory of Bovine Disease Control in Northeast China, Ministry of Agriculture and Rural Affair, College of Animal Science and Veterinary Medicine, Heilongjiang Bayi Agricultural University, Daqing 163319, China; 2School of Public Health, Shantou University, Shantou 515041, China; 3State Key Laboratory of Pathogen and Biosecurity, Beijing Institute of Microbiology and Epidemiology, Beijing 100071, China

**Keywords:** tick, microbiota, microbial interactions, symbiont bacteria, 16S rRNA gene

## Abstract

**Simple Summary:**

Ticks pose a threat to humans and animals. However, microbial interactions on ticks are underappreciated. Complex microbial interactions shape microbial communities. There is limited information about microbial interactions on ticks, including *Dermacentor silvarum*, *Ixodes persulcatus*, and *Haemaphysalis concinna*. This study evaluated the microbial community composition of these three species and the microbial interactions based on pairwise interactions. The results reveal that the bacterial richness and microbiota structures of the three tick species are significantly different, and the bacterial richness of all ticks decreased significantly after they became engorged. A substantial expansion of the list of bacterial interactions was observed in ticks.

**Abstract:**

Ticks carry and transmit a variety of pathogens, which are very harmful to humans and animals. To characterize the microbial interactions in ticks, we analysed the microbiota of the hard ticks, *Dermacentor silvarum*, *Ixodes persulcatus*, and *Haemaphysalis concinna,* using 16S rRNA, showing that microbial interactions are underappreciated in terms of shaping arthropod microbiomes. The results show that the bacterial richness and microbiota structures of these three tick species had significant differences. Interestingly, the bacterial richness (Chao1 index) of all ticks decreased significantly after they became engorged. All the operational taxonomic units (OTUs) were assigned to 26 phyla, 67 classes, 159 orders, 279 families, and 627 genera. Microbial interactions in *D. silvarum* demonstrated more connections than in *I. persulcatus* and *H. concinna.* Bacteria with a high abundance were not important families in microbial interactions. Positive interactions of Bacteroidaceae and F_Solibacteraceae Subgroup 3 with other bacterial families were detected in all nine groups of ticks. This study provides an overview of the microbiota structure and interactions of three tick species and improves our understanding of the role of the microbiota in tick physiology and vector capacity, thus being conducive to providing basic data for the prevention of ticks and tick-borne diseases.

## 1. Introduction

In blood-sucking arthropods, the microbiome can vary widely [1,2,3,4]. Globally, ticks pose a threat to animals and humans. The biological characteristics of most microorganisms and their effects on ticks have not yet been explored and are typically neglected. However, such microorganisms have a variety of harmful, neutral, and beneficial effects on host ticks and play a variety of roles in health, nutrition, adaptation, development, reproduction, defence against environmental stress, and immunity [5]. In addition to pathogenic microorganisms, non-pathogenic microorganisms may be involved in the transmission of tick-borne pathogens, which may impact animal and human health in numerous ways [6]. Despite the growing knowledge about the factors that influence the composition and abundance of the microbiota, there are still many unanswered questions regarding microbiota in ticks. Currently, there is no consensus about which factor is conducive to the characterization of tick microbiota composition, and there is a lack of understanding of the different relationships between tick microbiota [7].

More than 900 valid species of tick are known worldwide, of which 124 species are found in China [8]. The growing number of tick bites each year poses an escalating risk of tick-borne diseases. *Dermacentor silvarum, Ixodes persulcatus,* and *Haemaphysalis concinna* (Acari: Ixodidae) are abundant and epidemiologically important tick species in China. All three tick species have a wide range of hosts and different host preferences at different life stages, and all have been reported to bite humans and animals [8]. Thus, the structure of their microbiota should be studied carefully. In addition to being the principal vector for the Lyme disease pathogen, *Borreliella burgdorferi*, these three species carry numerous other medically important pathogens: *Anaplasma* spp., *Babesia* spp., *Ehrlichia* spp., *Rickettsia* spp., and tick-borne encephalitis virus [9]. Moreover, alongside the tick-borne pathogens, ticks also harbour non-pathogenic microorganisms such as endosymbiotic, commensal, mutualistic, and parasitic bacteria [10]. There is evidence that tick microbial communities may affect their vectorial capacities.

Some representative bacterial genera may behave as subtle symbionts engaged in intricate interactions with ticks. In certain ticks, a large number of vertically transmitted endosymbionts are found, which may play a trophic role by providing essential nutrients to the host [11,12,13]. These nonpathogenic bacteria may interact with a variety of tick-borne pathogens, including *Anaplasma marginale*, *B. burgdorferi*, and other *Rickettsia* species [14,15,16].

Due to the recent resurgence of tick-borne infectious diseases worldwide, research on tick microbial composition is crucial. To understand the relationship between microbes and hosts, it is important to unravel the mystery of the composition of microbial communities. However, to date, there is limited information on the microbial interactions of these three species of ticks. In this study, we investigated the structure and composition of *I. persulcatus, H. concinna*, and *D. silvarum* collected from the field as major tick vector communities to better understand the forces shaping tick microbiota. In order to gain a better understanding of tick microbial interactions, we analysed microbial interaction networks. This study examined the composition of tick microbial communities and investigated the symbiotic and antagonistic relationships between microorganisms, which are conducive to providing basic data for the prevention of ticks and tick-borne diseases.

## 2. Materials and Methods

### 2.1. Tick Collection and DNA Extraction

Three tick species were collected in the forest of Harbin, Jiamusi, and Great Khingan in Heilongjiang Province of China from March to June 2021, according to their active areas and months. Unfed female and male ticks were collected from the wild forest using dragging, and engorged ticks were collected from cattle in the same locations. A total of 2464 ticks were collected, including 2375 unfed and 89 engorged ticks. Among them, 85 (22 engorged) ticks were used in the study, including 27 (9 engorged) *I. persulcatus*, 36 (9 engorged) *D. silvarum*, and 22 (4 engorged) *H. concinna*. All ticks were adults, and the detailed sample information is shown in Table 1. The tick species were identified by their morphological characteristics and molecular data based on mitochondrial 16S rDNA sequences [10,17]. In order to clean the microorganisms from the surface of ticks to reduce environmental contamination, a method involving bleach was used [18]. Briefly, a 1% commercial bleach solution was used on the first batch of ticks to process them for 30 s, followed by three rinses with DNA-free water, each lasting 1 min. The cleaned ticks were preserved intact at −20 °C for DNA extraction.

The ticks were analysed individually. Before DNA extraction, the ticks were dried on sterile filter paper and, under liquid nitrogen, ground with steel balls for five minutes. A DNA Mini kit (QiAGEN, Shanghai, China) was used to isolate DNA. Nucleic acid was stored at −80 °C until use.

### 2.2. High-Throughput Sequencing and Bioinformatics Analysis

As for the tick microbiota, the ribosome 16S V3–V4 region is commonly used for accurate taxonomic differentiation [16,19]. Therefore, according to previous reports, the V3–V4 region of the bacterial 16S rRNA gene was amplified. We sequenced the DNA using an Illumina Hiseq 2500 platform following a standard protocol. The NCBI accession number for the raw sequencing data reported here is PRJNA849745.

At 97% sequence identity, all effective tags clustered into operational taxonomic units (OTUs) [20]. We used the Mothur method and the SSUrRNA [21] database of SILVA [22] to conduct species annotation analysis (setting a threshold of 0.8–1), acquire information on taxonomy, and, at each classification level (kingdom, phylum, class, order, family, genus, and species), identify the community composition of each sample. MUSCLE 3.8.31 [23] software was used for multiple sequence alignment, and all OTUs were obtained on behalf of the sequence of the system. The statistical analysis involved in this work was implemented using the R 4.1.3 (http://www.r-project.org, accessed on 10 May 2022) platform. The Shannon and Chao1 indexes of samples were calculated using the ‘diversity’ and ‘estimateR’ functions in the vegan (v.2.6-2) package [24] (https://cran.r-project.org/package=vegan, accessed on 8 June 2022), respectively. The Bray–Curtis distance between samples was calculated using the ‘vegdist’ function in the vegan package [25]. The ‘cmdscale’ function was used to realize principal coordinate analysis (PCoA), and the results of PCoA were visualized by the scatterplot 3D package [26] (https://CRAN.R-project.org/package=scatterplot3d, accessed on 11 June 2022). The ‘adonis’ function in the vegan package was used for the permutational multivariate analysis of variance (PERMANOVA) analysis. The Linear discriminant analysis effect size (LEfSe) was used to find the genera of bacteria that differed between groups and was performed on the Galaxy online platform (https://huttenhower.sph.harvard.edu/galaxy, accessed on 21 June 2022). Comparisons of the differences in microbial diversity and classification level between the groups were obtained by Wilcoxon rank-sum test.

### 2.3. Detection of Complex Interaction Patterns

Microbial interaction analysis was conducted independently on each group of ticks. The Spearman correlation coefficient was used to evaluate the correlation between bacterial families based on the relative abundance spectrum [27]. Only strong inter-correlations (*p* > 0.01) were retained in order to clearly show associations between families. The visualization of correlations was achieved using the R package “ggplot2” and Gephi software (https://gephi.org/, accessed on 24 June 2022) [28,29].

## 3. Results

### 3.1. Microbiota Diversity

A total of 85 (22 engorged) adult ticks from three species were sequenced for the V3-V4 region of the 16S rRNA gene. After quality control, 8,869,556 clean reads were obtained. A total of 1810 OTUs were obtained through splicing and 97% identity threshold clustering (Appendix A). Based on our rarefaction curve analysis, we were able to detect all OTUs in tick samples using our sequencing depth.

The α-diversity of the microbiota in the samples was demonstrated in two ways: using the Shannon index and Chao1 index. The diversity of the microbial communities varied significantly in *H. concinna* compared with *D. silvarum* and *I. persulcatus* in terms of the OTU level (Figure 1A). Female ticks from either the engorged or unfed samples had significantly higher diversity than male ticks within *D. silvarum* (Figure 1B). No difference was observed between groups in *H. concinna*, whereas *I. persulcatus* females had significantly lower diversity than males (Figure 1B). There were no significant differences in terms of male tick richness in any of the species (Figure 1B). Something more interesting was found when exploring the Chao1 index of the samples. Among the three tick species, *H. concinna* had the lowest Chao1 index, which differed from the trend reflected in the Shannon index (Figure 1C). This indicates that low-abundance bacterial OTUs are rare in *H. concinna,* and the number of OTUs was lower in *H. concinna* than in the other two tick species. In addition, we compared the samples of different sexes and feed stages in the same tick species (Figure 1D) and found that when the ticks were engorged, the Chao1 index significantly decreased.

### 3.2. Factors That Influence Microbiota Community Structure

To determine how sexes and feed stages influence the microbial composition, we compared the β-diversity of the microbiota of male and female (unfed or engorged) individuals within each tick species. All *H. concinna* samples showed a discrete distribution. Regarding *I. spersulcatu*, the microbial composition of engorged female samples was significantly different. The male and unfed female samples were clustered in different locations, and there was no significant difference within the group (Figure 2A). The β-diversity of males was more divergent in all three tick species but was not significantly different from that of engorged females; unfed females showed significant differences (Figure 2B).

### 3.3. Common and Differentially Abundant Bacteria between and within Tick Species

The data were examined to determine if tick species share or possess unique bacterial genera. The bacteria were largely common in all species. No matter which group, *I. spersulcatus* had more unique bacterial genera, but among all of the engorged female samples, *D. silvarum* had the most specific bacterial genera. Similarly, we analysed the genera of bacteria that were unique or shared among females, males, and engorged females. Unfed females had the most unique bacterial genera in samples of *D. silvarum* and *I. spersulcatus*, whereas engorged females had the most unique bacterial genera in samples of *H. concinna* (Figure 3A).

All of the OTUs were assigned to 26 phyla, 67 classes, 159 orders, 279 families, and 627 genera, the detailed information of which is presented in Appendix A. At the phylum level, Proteobacteria and Firmicutes were the dominant phyla in all ticks. Additionally, Bacteroidetes, Actinobacteria, and Acidobacteria were the main symbiotic bacteria in all ticks (Figure 3B). In *D. silvarum*, the relative abundance of Proteobacteria in all samples was higher than that in Firmicutes, and the relative abundance of Proteobacteria in the female samples was higher than that in the male samples. The relative abundance of Proteobacteria showed a decreasing trend after the ticks were filled with blood. In contrast, the relative abundance of Proteobacteria decreased after *I. persulcatus* ticks were filled with blood. The bacteria of different phyla in *H. concinna* showed a relatively stable state, consistent with the α-diversity results (Figure 3B).

The histogram of sample distribution at the family level showed that bacteria in all samples were mainly distributed in Lactobacillaceae, Veillonellaceae, Streptococcaceae, Ruminococcaceae, Bacteroidaceae, Muribaculaceae, and Rickettsiaceae. Enterobacteriaceae was the most common family. The relative abundance of Enterobacteriaceae increased, and that of Rickettsiae decreased after *D. silvarum* was engorged. Different from *D. silvarum*, the relative abundance of Rickettsiae increased significantly after *I. persulcatus* was engorged. The relative abundance of Lachnospiraceae increased after *H. concinna* was engorged (Figure 3C).

According to the genus-level analysis of sequencing data, the microbiota of the three species were dominated by *Lactobacillus, Rickettsia, Megamonas,* and *Escherichia-Shigella. Arsenophonus* was a highly abundant genus in *D. silvarum*. The relative abundance of *Rickettsia* was higher in unfed *D. silvarum*. The relative abundances of *Arsenophonus* and *Pseudomonas* increased after the ticks were engorged. The relative abundance of *Lactobacillus* was higher in unfed *I. persulcatus*. The relative abundance of *Rickettsia* and *Pseudomonas* increased, whereas the relative abundance of Lactobacillus decreased when the ticks were engorged. No significant changes were observed in the sex or feed stage of *H. concinna* (Figure 3D).

To detect the microbial signatures in each factor, LEfSe was conducted. When *H. concinna* was compared with the other two species, the microbiota showed the greatest differences, in agreement with the results of our β-diversity analysis (Appendix A). Different bacterial genera were discovered between species regardless of feed stage or sex. It seems that feed stage and sex are not very influential on these specific bacterial genera. For example, *Megamonas* and *Serratia* were more abundant in *H. concinna* than in *I. persulcatus* and *D. silvarum,* regardless of sex (Appendix A). Based on differences in abundance within the same species, we found infection gradients between sexes and feed stages. For example, *Coxiella* and *Arsenophonus* were found in higher abundance in female *D. silvarum* than in male ticks and engorged ticks (Appendix A). We were able to determine the bacterial abundance of specific microbes across species by comparing differential abundances within a group. For example, in *I. persulcatus* and *D. silvarum*, the abundance of *Serratia* and *Macromonas* was higher in unfed females than in males and engorged females (Appendix A).

### 3.4. Microbial Interaction within Ticks

In order to identify potential patterns of microbial interaction, histograms and networks of microbial interactions were created based on data from 16S rRNA sequencing. The Computational model used presence/absence and relative abundance data to identify pairwise relationships between bacteria families. In general, microbial interactions with *D. silvarum* had more connections than with *I. persulcatus* and *H. concinna* (Figure 4, Appendix A). All networks showed antagonism and synergy interaction patterns for all tick species except for the male *D. silvarum*. The number of positive interactions was higher than that of negative interactions (Appendix A). We were able to identify taxa within these interactions that appeared to be important to the overall structure of the microbiota. Interestingly, we found that bacteria with high abundance were not important families in network interactions. Only three of the five most abundant families showed interactions. Rickettsiaceae and Lactobacillaceae did not show interactions in all samples. Positive interactions of Bacteroidaceae and F_Solibacteraceae Subgroup 3 with other bacterial families were detected in all groups (Figure 4).

## 4. Discussion

In order to gain a deeper understanding of the factors that influence the tick microbiota, we sequenced the microbiota of field-caught *I. persulcatus*, *D. silvarum*, and *H. concinna* adult ticks, which are common species in China. Evidently, the factors that make up the microbiota are complex. The Shannon index showed significant differences between the three species of ticks in terms of their microbiota diversity. The total bacterial diversity was similar between different sexes and feed stages in *H. concinna*, but it was different in *I. persulcatus* and *D. silvarum*. The tick species was also found to affect bacterial diversity, as in other studies [30,31]. The Chao1 index significantly decreased when the ticks were engorged, which suggests that some low-abundance bacteria were killed or expelled after a tick was fed. Some bacteria that initially inhabit the blood of animals or humans enter ticks through blood-feeding. Ticks have complex life cycles, and the off-host period is long; therefore, the microbiota inhabiting ticks could be shaped by the environment and, through interactions with blood meals, have an influence on the tick microbiota. Some studies of tick microbes have reported similar results [32,33,34]. When possible, blood feeding activates the immune system of the tick or insect, inhibited by the immune deficiency (IMD) pathway, which results in antimicrobial peptides (AMP) being expressed and bacteria surviving. The microorganisms that can adapt to this changed microenvironment have a greatly increased abundance in the gut bacteria. The diversity of microorganisms was significantly reduced due to some species being eliminated or greatly reduced [16]. However, the groupings of individuals from *H. concinna* showed less variation when comparing sexes and feed stages. At the same time, these results indicate that the bacterial diversity of male individuals was higher than that of female individuals in *D. silvarum* and *I. persulcatus*. There is no doubt that tick sex influences the diversity of microbial communities. We found that male ticks have a more diverse bacterial microbiota than female ticks in *D. silvarum* and *I. persulcatus*, which was consistent with other studies. For example, male *Dermacentor reticulatus* and *Dermacentor marginatus* have relatively richer and more diverse microbiotas than female ticks [35]. Additionally, similar results were found in *Haemaphysalis longicornis* [35,36], *Rhipicephalus* (Boophilus) *microplus* [37], and *Rhipicephalus turanicus* [38]. However, there was no significant difference in *H. concinna* in our study. The reason for this finding needs to be determined by further analysis.

Our findings reveal that the phylum Proteobacteria is the most dominant tick microbiota, followed by Firmicutes, Bacteroidetes, and Acinetobacter, which is consistent with previous research in ticks [31,38,39]. Several bacterial families were the same in different tick species, such as Enterobacteriaceae, Lactobacillaceae, Ruminococcaceae, etc., which was similar to findings in other tick species [39,40]. New microbial control strategies for ticks could use these common taxa as candidate bacteria, especially those causing infection at high abundances. Additionally, these core bacterial florae play indispensable roles in the growth and development of arthropods [41]. In comparison with *D. silvarum* and *I. persulcatus*, *H. concinna* had more differentially abundant genera, which explains the greater divergence of its microbiota. We observed a large number of different bacterial genera in ticks, but the genera that have been reported to have important effects on tick growth and development were not found in all tick samples; the reason may be that other symbiotic bacteria perform the same function in different tick species. For example, the Coxiella-LE genome was shown to encode the major pathways responsible for the synthesis of B vitamins, such as biotin (B7 vitamin), folic acid (B9), riboflavin (B2), and their cofactors, which are not usually obtainable in sufficient quantities from a uniquely blood-based diet [42]. However, Coxiella-LE was not detected in all the samples in this study. Other bacteria that code for the same gene also constitute an area for future research.

It is unknown how ticks interact with microorganisms. The majority of evidence for microbial interactions within blood-sucking arthropods comes from *Wolbachia* and its colonies [43,44,45,46] or transmitted symbionts in other arthropods [14,47,48]. Therefore, we are relatively familiar with inherited symbionts, whereas we are less familiar with the magnitude of interactions between the microbes found in arthropod guts. Our analysis identified synergistic and antagonistic interactions, which substantially increased the number of bacterial interactions observed in ticks. The α-diversity of the microbiota may explain the differences observed in the network structure. More complex networks generally had more OTUs. Network analysis using the entire dataset revealed that most of the detected interactions among members of the *D. silvarum* microbiota were positive, whereas those in other tick species were different. Taxa with positive associations are often interpreted as bacteria that perform similar or complementary functions [49,50]. Negative associations may also reflect interactions such as competition and niche partitioning. Accordingly, the majority of correlations being positive indicates that *D. silvarum* microbial communities perform similar or complementary functions and favour mutualism; this is not the case with the other two species.

Many factors may influence the microbiota network structure. In particular, the microbial networks could differ in tissues such as the salivary glands, germline, and gut. For example, microbial network analysis from the human microbiota project revealed that microbial networks were different depending on the site of the body [51,52]. Within the same individual, the microbiota of the reproductive organs was more diverse than that of the gut and salivary glands. Salivary glands, mostly in Anophelines, showed higher diversity indices when compared with the guts, similar to that reported in *Anophelines Culicifacies* [53]. Here, the whole tick was used for sampling, whereas bacteria can reside in different organs of a tick. Although there is no direct interaction, it is possible for co-occurrence at the tick level to result in indirect interactions, such as the host immune system. In the future, the analysis of distinct tissues may uncover further information.

## 5. Conclusions

The bacterial richness and microbiota structures of *D. silvarum*, *H. concinna*, and *I. persulcatus* were significantly different, and the bacterial richness of all ticks decreased significantly after they became engorged. There were synergistic or antagonistic relationships among co-occurring bacteria of different tick species, which are conducive to providing basic data for the prevention of ticks and tick-borne diseases.

## Figures and Tables

**Figure 1 animals-12-03237-f001:**
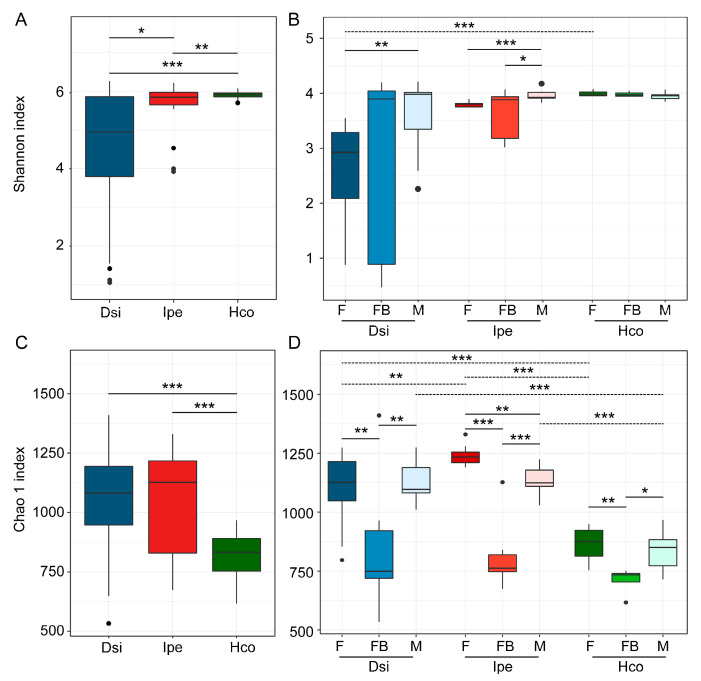
Microbial diversity and richness in samples. (**A**,**B**) Shannon diversity indices at the OTU level for all tick species (**A**) or for each group within a species (**B**). (**C**,**D**) Chao1 indices at the OTU level for all tick species (**C**) or for each group within a species (**D**). The Wilcoxon rank test was used to determine significance (* *p* < 0.05, ** *p* < 0.01, *** *p* < 0.001) within species (full line) or groups (dotted line). Notes: F, female; FB, full of blood (engorged); M, male; Dsi, *Dermacentor silvarum*; Ipe, *Ixodes spersulcatus*; Hco, *Haemaphysalis concinna*.

**Figure 2 animals-12-03237-f002:**
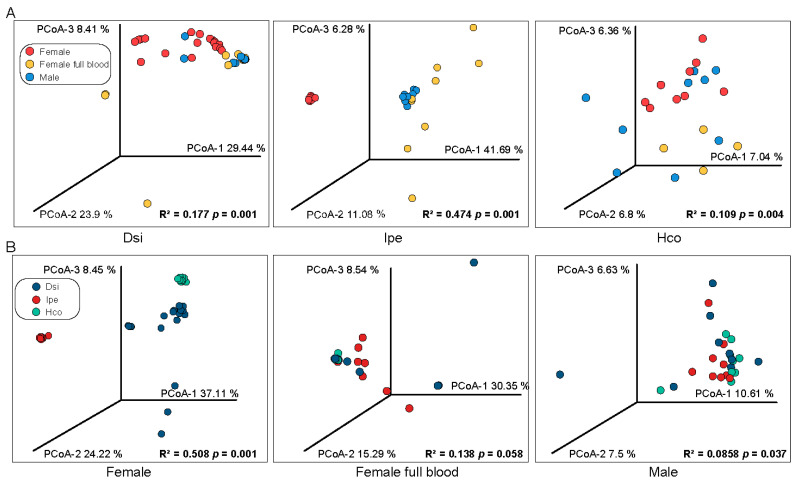
PCA analyses for detecting similarities between different samples. Principal coordinate analysis (OTU level) using Bray–Curtis dissimilarity, comparing identified OTUs within a group (**A**) or species (**B**). Notes: The R^2^ and *p* values generated by the PERMANOVA analysis are shown in the lower right of the plot. Dsi, *Dermacentor silvarum*; Ipe, *Ixodes spersulcatus*; Hco, *Haemaphysalis concinna*.

**Figure 3 animals-12-03237-f003:**
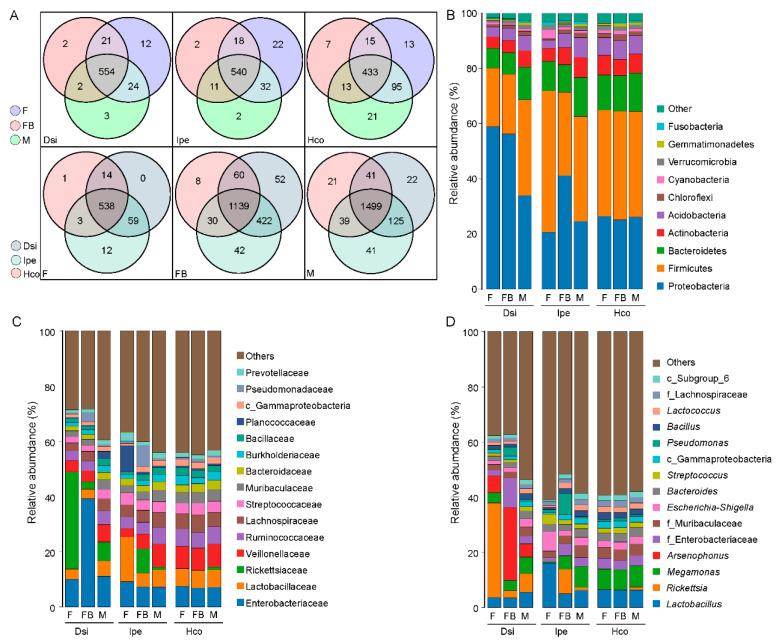
Common and differentially abundant bacteria (different level) within ticks. Venn diagram showing number of common bacterial genera between tick groups and species (**A**). Histogram of species distribution at different taxonomic levels (**B**–**D**). Notes: F, female; FB, full of blood (engorged); M, male; Dsi, *Dermacentor silvarum*; Ipe, *Ixodes spersulcatus*; Hco, *Haemaphysalis concinna*.

**Figure 4 animals-12-03237-f004:**
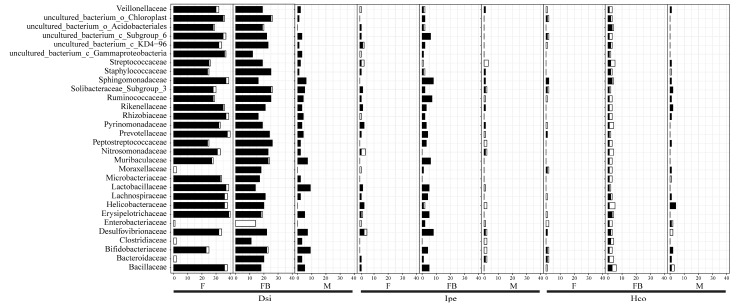
Microbial interactions within ticks. The top 30 bacterial families are included in the chart. White represents co-exclusion/negative correlation, black represents co-occurrence/positive correlation interactions. Notes: F, female; FB, full of blood (engorged); M, male; Dsi, *Dermacentor silvarum*; Ipe, *Ixodes spersulcatus*; Hco, *Haemaphysalis concinna*.

**Table 1 animals-12-03237-t001:** The information of the ticks collected in this study.

Samples ID	Number	Species	Sex	Feed Stage
D1-D18	18	*D. silvarum*	female	unfed
D19-D27	9	*D. silvarum*	female	engorged
D28-D36	9	*D. silvarum*	male	-
H1-H9	9	*H. concinna*	female	unfed
H10-H13	4	*H. concinna*	female	engorged
H14-H22	9	*H. concinna*	male	-
I1-I9	9	*I. persuleatus*	female	unfed
I10-I18	9	*I. persuleatus*	female	engorged
I19-I27	9	*I. persuleatus*	male	-

## Data Availability

The data presented in this study are openly available in NCBI at https://www.ncbi.nlm.nih.gov/bioproject/PRJNA849745, accessed on 8 June 2022, accession number PRJNA849745.

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
