# Peer review of "Microbiota Community Structure and Interaction Networks within Dermacentor silvarum, Ixodes persulcatus, and Haemaphysalis concinna"

_animals, 2022, doi:10.3390/ani12233237_

Round 1

Reviewer 1 Report

It is an interesting paper reporting the microbiota of ticks of Dermacentor silvarum, Ixodes persulcatus, and Haemaphysalis concinna. It has merits for publication in Animals once the following issues are addressed:

Line 31-32: Rewrite to clarify. What does ‘expansion’ mean?

Line 69: Borreliella burgdorferi or Borrelia burgdorferi?

Line 94: How many individuals of ticks were collected and how many of them were used for analyses? How were the samples selected? Collection sites and dates are not mentioned. Were they collected at the same time?

Line 97: In view of the high frequency of cryptic species in ticks, were the IDs confirmed molecularly?

Line 99-100: Justify why bleach was used over other surface sterilization methods

Line 106: Were control groups included to rule out lab contaminants?

Line 108: Justify further why V3-V4 region was selected over other regions

Line 139: How many males, females, engorged samples were used?

Discussion: Please discuss if the microorganisms detected from the engorged specimens originated from their hosts

Reviewer 2 Report

The manuscript concerns microbial community structures in three ixodid species, important TBP vectors in China. The study seems to be well planned, the tick sampling is sufficient for this study, sequence data and their analysis do not raise concerns. The results obtained are valuable and bring to the knowledge. Unfortunately, the paper is written very poorly in English, in many parts it is not clear what the Authors mean. Also, the results are poorly presented - there are really a number of results, and they need to be adequately described. I don't even judge the discussion by the fact that in lines 263-266 the authors pasted/forgot to remove the reviewer's comments, perhaps from a previous journal to which the paper was sent previously.

Below I provide other comments, which may help the authors to prepare a revised version of the manuscript. Most of them relate to the writing and content of the text, but they are not all comments. The paper should be thoroughly improved in many other places as well:

1. Abstract is much simpler than simple summary; it is more general and presents less results.

2. Some statements in the introduction require literature references, e.g., those regarding “no consensus about which factor is conducive to the characterization of tick microbiota composition” (lines 53-55).

3. “high-throughput sequencing of 16S rRNA amplicons identified inter-62 actions in both Drosophila and mosquitoes” (lines 62-63) – A mental shortcut leading to erroneous information (the effect of flies on mosquitoes and vice versa was not studied, only the effect of changes in the microbiome on these organisms).

4. Line 72 – it is hard to agree that only I. persulcatus harbor also non-pathogenic bacteria.

5. The introduction is vague; it should include information on why the focus is on these tick species (they have different host ranges? different life cycles?), instead of trivialities about “the mystery” of the tick’s microbiome composition.

6. The main text must include information about the analyzed material: how many, what stages, where caught, at what time - month and year - all this is important data that can be very useful to readers in the future. Such information cannot be hidden in supplementary material.

7. Use “host-seeking” or “unfed” instead “free” when refer to females caught by flagging.

8. All bioinformatics tools should be well cited, this is often missing, such as for Cytoscape, but not only.

9. Methodology is unnecessarily given in the results (e.g., lines 145-146, 152-153, but not only).

10. All abbreviations in the drawings should be explained in the caption, including those that seem obvious (F, FB, M, etc.).

11. The colors in Fig. 1c do not correspond to the markings in the other figures (a, b, and d).

12. PCA results are not correctly interpreted, males do not differ significantly in microbiome composition (fig. 2A). The authors seem to have confused the species diversity of the microbiome with how much the microbiota have differed between individuals (or grouped based on the same bacterial taxa). (Microbial diversity has been shown in fig. 3).

13. Results like “All the OTUs were assigned to 26 phyla, 67 classes, 159 orders, 279 families, and 627 genera” actually says nothing. However, to which result the statement “Additionally, in the female engored blood samples, 52, 42, and eight were in D. silvarum, I. persulcatus, and H. concinna, respectively” refers? And what does it mean?

14. Fig. 4 is unclear, even on enlarged pdf document. It is better to focus only on those relationships that are statistically significant and are discussed and shown in the abstract. Then the figure(s) would be larger and more readable. Now the diagrams are only colorful and pretty, but uninformative. 

15. What do the colors other than green and red in Fig. 4 mean?

Round 2

Reviewer 2 Report

The authors have significantly improved the manuscript, but there are still some issues that should be addressed before accepting the paper:

line 20 "allticks" should be with space

line 91 "from cattle in the same cities" (sites? locations?)

line 96 "molecular biology" should be "molecular data"

line 97 "Information including sex and feed stage was recorded synchronously" should be removed

line 98 "In order to clean the surface of the microorganisms" should be "In order to clean the microorganisms from the surface of ticks" (Now it is unclear and seems to be about cleaning the surface of microorganisms).

line 103 "The tick samples used for high-throughput sequencing are individual samples." should be "The ticks were analyzed individually."

line 104-105, the sentence should be "Before DNA extraction, the ticks were dried on sterile filter paper and, under liquid nitrogen, ground with steel balls for five minutes."

line 110 "Sequencing" should be lowercase

Fig. 4 should be in high-resolution in the final version of the paperline

line 320 "Synergistic" should be lowercase.
